# Multiscale Simulation of Laser-Based Direct Energy Deposition (DED-LB/M) Using Powder Feedstock for Surface Repair of Aluminum Alloy

**DOI:** 10.3390/ma17143559

**Published:** 2024-07-18

**Authors:** Xiaosong Zhou, Zhenchao Pei, Zhongkui Liu, Lihang Yang, Yubo Yin, Yinfeng He, Quan Wu, Yi Nie

**Affiliations:** 1School of Mechanical and Electrical Engineering, Guizhou Normal University, Guiyang 550001, China; ivzhouxiaosong@hotmail.com (X.Z.); pzc330230313@163.com (Z.P.); wu_quan@gznu.edu.cn (Q.W.); 2Nottingham Ningbo China Beacons of Excellence Research and Innovation Institute, University of Nottingham Ningbo China, Ningbo 315100, China; zhongkui.liu@nottingham.edu.cn (Z.L.); lihang.yang2@nottingham.edu.cn (L.Y.); yinfeng.he@nottingham.edu.cn (Y.H.); 3Faculty of Science and Engineering, University of Nottingham Ningbo China, Ningbo 315100, China; 4Faculty of Engineering, University of Nottingham, Nottingham NG7 2RD, UK

**Keywords:** laser direct energy deposition, surface repair, aluminum alloy, multiscale simulation, molten pool, thermal stress

## Abstract

Laser-based direct energy deposition (DED-LB/M) has been a promising option for the surface repair of structural aluminum alloys due to the advantages it offers, including a small heat-affected zone, high forming accuracy, and adjustable deposition materials. However, the unequal powder particle size during powder-based DED-LB/M can cause unstable flow and an uneven material flow rate per unit of time, resulting in defects such as pores, uneven deposition layers, and cracks. This paper presents a multiscale, multiphysics numerical model to investigate the underlying mechanism during the powder-based DED-LB/M surface repair process. First, the worn surfaces of aluminum alloy components with different flaw shapes and sizes were characterized and modeled. The fluid flow of the molten pool during material deposition on the worn surfaces was then investigated using a model that coupled the mesoscale discrete element method (DEM) and the finite volume method (FVM). The effect of flaw size and powder supply quantity on the evolution of the molten pool temperature, morphology, and dynamics was evaluated. The rapid heat transfer and variation in thermal stress during the multilayer DED-LB/M process were further illustrated using a macroscale thermomechanical model. The maximum stress was observed and compared with the yield stress of the adopted material, and no relative sliding was observed between deposited layers and substrate components.

## 1. Introduction

Aluminum alloys are a crucial group of materials in the aerospace and automotive fields owing to their outstanding specific stiffness and strength [1,2]. However, flaws such as wear, cracks, and holes are unavoidable after a continuous external load, which undermines their reliability in servicing conditions and can sometimes result in fatal fractures. Compared with the cost of direct disposal or replacement of whole parts, surface repairs of the compromised parts has always been an attractive alternative choice. Traditional surface repair methods, including casting and forging, rely on producing an entire replacement body at local places, which is time-consuming and expensive [3]. The alternative solution, e.g., metal patching on the damaged surface, can only be a temporary backup in an emergency with the sacrifice of property consistency and the introduction of extra weight [4].

Laser-based direct energy deposition (DED-LB/M), often also referred to as laser metal deposition (LMD), laser cladding (LC), or laser welding (LW), utilizes a laser source to heat the powder or wire feedstock material to form a molten pool deposited and cooled down on the substrate, and is a promising choice for the surface repair of compromised parts [3]. However, defects such as pores, unevenly deposited layers, and cracks are always present during the powder-based DED-LB/M process due to the rapid heating and cooling and the non-uniform powder particle stream. The complex physical phenomenon during the powder-based DED-LB/M process can be observed by an in situ high-speed camera [5] or via X-ray imaging [6] for molten dimensions. The heat or temperature field has also been directly [7] or indirectly [8] captured by a suitable monitoring setup. Nevertheless, such expensive testing facilities only provide researchers with data regarding the ongoing phenomenon and with limited information on the underlying physics. Computational simulation provides an insightful method to reveal the complicated physics inside the molten pool and resolve the mechanism that produces defects during the powder-based DED-LB/M process [9].

To investigate the fluid flow inside the molten pool during the powder-based DED-LB/M process, the challenge is to suitably capture the interaction between the discrete-based non-uniform powder feeding stream and the continuum-based molten pool. Wang et al. [10,11] and Bayat et al. [12] applied the discrete phase method (DPM) to simplify the supplied metal powder as Lagrangian particles and restored the physical mass, thermal energy, and kinetic energy exchange during the powder-based DED-LB/M process by using FLOW-3D software (https://www.flow3d.com/). The physical shape of powder particles and the interaction between laser and metal powder are neglected by setting the temperature of the deposited Lagrangian particles before entering the molten pool as their liquidus temperature. Sun et al. [13] coupled DPM with the volume of fluid (VOF) to capture the interactions between the powder stream and molten pool and adopted the enthalpy–porosity method to account for phase change. By assigning the powder particles to their liquidus temperature, as granted by the nature of the coupled method, the powder particles can transform from a discrete phase to a continuous phase when they are injected at the local molten pool surface. The fluid in the regions where the powder was injected was captured by a downward flow and a reduction in temperature. Good agreements between the simulation model and experiments were obtained.

Considering the dynamic behaviors, physical size, and shape of the delivered powder, it is appropriate and more convincing to employ the discrete element method (DEM) to model the powder-based DED-LB/M process [14]. Aggarwal et al. [15] developed the coupled DEM and finite volume method (FVM) to investigate the interactions between the molten pool and the impacting powder particles. It was observed that the momentum introduced by the impacting powder particles outweighs the Marangoni effect, thereby stimulating the melting of the metal in the molten pool, which is in contrast to past findings. The obtained morphology of the molten pool and temperature fields were verified by the conducted experimental measurements. Khairallah et al. [16] employed the ALE3d multiphysics code to develop a high-fidelity mesoscale numerical model for the powder-based DED-LB/M process. It was predicted that the laser absorptivity would stay around 0.4 regardless of variations in process parameters. Although the high-fidelity nature of mesoscale molten pool simulation contributes to its accurate spatial and temporal results, the huge computational burden limits the situations in which it can be applied.

In addition to the above models of fluid flow, the rapid heat transfer and variation in thermal stress accompanied by the molten pool have also been studied. Srivastava et al. [17] developed a thermomechanical model for the arc-based DED process to quantify the residual stresses and deformations of the produced components. Stender et al. [18] employed a heat source to heat the activated elements of composite materials at each time step. The temperature and topology of the materials are then transferred to a solid mechanics analysis, allowing for the computation of displacement and stress fields. Li et al. [19,20] focused on calculating the thermal history at the macroscopic scale and directly used it for thermomechanical resolution without considering local thermal histories. Bresson et al. [21] systematically defined the spatiotemporal boundaries at each layer and the modeling strategies of the heat source’s initial and boundary conditions. This approach was used to determine the position of porosity and calculate the thermal history, which is crucial for understanding the formation of residual stresses and deformation generated during the DED-LB/M process.

Existing computational studies on DED-LB/M have been primarily focused on the additive manufacturing (AM) aspect to layer material up, with limited attention on the surface repair aspect to restore flaws such as holes and cracks. This paper aims to fill the research gap in the multiscale and multiphysics simulation of the coaxial-powder-based DED-LB/M technology for surface repair or remanufacturing. A mesoscale coupled DEM-FVM model and a macroscale thermomechanical coupled model are established to illustrate the temperature field, velocity field, and pressure field variations in the molten pool, analyze the morphology evolution of the molten pool, evaluate the repair effectiveness of different-sized surface defects, and reveal the formation mechanisms of pores or uneven deposited layers and of residual stress and deformation during the DED-LB/M surface repair process.

## 2. Physics during the Powder-Based DED-LB/M Process

The physical phenomenon involved in the powder-based DED-LB/M process starts with the laser heat source (e.g., a Gaussian laser beam)-induced thermal radiation on the delivered metal powder and the rapid melting and formation of a molten pool, shown in in Figure 1. Within the molten pool, thermal radiation, heat diffusion, and evaporative heat dissipation occur simultaneously. The molten pool and the surrounding air generate a dynamic flow of multiphase fluid. The flow of the molten pool is influenced by surface tension, mushy zone drag forces, and the Marangoni effect. As the laser scans over the region, the molten pool rapidly cools and solidifies, forming a deposit track. The stability of the molten pool during the powder-based DED-LB/M process affects the surface quality and internal structure of the deposited layer. Multiple deposit tracks form a deposit layer, and the deposit layers stack on top of each other to create the final deposit surface. During the powder-based DED-LB/M process, the material is built up layer by layer, with each layer being radiatively heated by the laser heat source. Heat conduction occurs within the material, resulting in rapid temperature increases and decreases. The material also undergoes repeated heating and cooling, leading to the formation of significant temperature gradients, thermal stresses, and thermal deformations within the material.

The physical processes involved in the interaction between laser and metal powder can be divided into four scales based on the desired research objectives. At the nanoscale or grain scale (<0.001–0.01 mm), the growth process of metal grains and the resulting internal grain structure can be studied based on the computed temperature field. At the mesoscale or powder scale (0.01–1.0 mm), a model considering particle morphology and size distribution can calculate the transportation and deposition processes of particles. By combining computational fluid dynamics with multiphysics analysis, the morphology and flow conditions of the molten pool can be studied, and the detailed morphology and internal porosity of the molten track can be predicted. The macroscale can be further divided into two levels: the scanning spacing scale (0.1–10 mm) and the part scale (1–250 mm). In terms of the scanning spacing scale, the deposition processes of individual scan tracks can be studied. Through thermomechanical coupling analysis, the evolution conditions, including temperature fields, stress fields, and deformation fields, during the deposition process can be obtained, allowing for the investigation of thermal stresses and deformations on the surface of the substrate part and between deposit layers. At the part scale (1–250 mm), the surface deposition process of large-sized components is considered. Similarly, with the aid of thermomechanical coupling analysis, the stress and deformation distribution of the parts, molten tracks, as well as between deposit layers during the powder-based DED-LB/M process can be studied.

This work will focus on the numerical simulation of the molten pool multiphysics fields at the mesoscale and a thermomechanical analysis at the macroscale during the surface repair of substrate components using the powder-based DED-LB/M technique. By accessing the numerical results, an evaluation based on the adopted process parameters will be presented.

## 3. Coupled DEM-FVM Modeling of the Molten Pool

Before establishing the numerical model for the molten pool flow process, it is necessary to reasonably simplify the complex physical process to ensure efficiency. This model will adopt incompressible Newtonian fluid laminar flow [22], neglect mass loss caused by metal vaporization [23], and disregard the influence of volume changes due to metal density variations [23]. The specific implementation process of the simulation is as follows.

### 3.1. Mathematical Modeling

The process of forming, flowing, and cooling a molten pool involves multiphysics fluid flow. The entire process is governed by the continuity equation, momentum conservation equation, and energy conservation equation. To fully account for the various physical factors affecting the molten pool during the powder-based DED-LB/M process, a two-phase fluid flow model is employed. Since the studied fluid is assumed to be incompressible, and the mass loss induced by gasification is ignored, the continuity equation in the computational domain becomes Equation (1):(1)∇·u=0

The finite volume method (FVM) is applied to track the metal–gas interface in the two-phase fluid flow model. The equations governing the law of element volume ratios for the metal and gas phases are Equations (2) and (3):(2)$$\frac{\partial \alpha_{1}}{\partial t}+\nabla \cdot \left(\alpha_{1}u\right)=0$$
(3)$$\alpha_{1}+\alpha_{2}=1$$
where t represents time, and $\alpha_{1}$ and $\alpha_{2}$ represent the elemental volume ratio of the gas phase and the metal phase, respectively. When $\alpha_{2}$ = 0, the region is fully occupied by the gas phase, while when $\alpha_{2}$ = 1, it means that the region is fully occupied by the metal phase. The $\alpha_{2}$ value ranges from 0 to 1 and accounts for the metal–gas interfacial region. According to the value of $\alpha_{2}$, the unit normal vector n at the gas–metal interface is derived as follows:(4)$$n=\frac{\nabla \alpha_{2}}{\left|\nabla \alpha_{2}\right|}$$

The curvature κ at the metal–gas interface is calculated as:(5)κ=−∇·n

The momentum conservation equation for the two-phase fluid flow domain is expressed as follows:(6)$$\bar{\rho }\frac{\partial u}{\partial t}+\bar{\rho }\left(u\cdot \nabla \right)u=-\nabla P+\bar{\mu }\nabla^{2}u+\bar{\rho }g+S_{mom}$$
among which:(7)$$\bar{\rho }=\alpha_{1}\rho_{1}+\alpha_{2}\rho_{2}$$
(8)$$\bar{\mu }=\alpha_{1}\mu_{1}+\alpha_{2}\mu_{2}$$
where $\bar{\rho }$, $\rho_{1}$, and $\rho_{2}$ are the density of the mixed gas and metal phase, the density of the gas phase, and the density of the metal phase, respectively. $\bar{\mu }$, $\mu_{1}$, and $\mu_{2}$ represent the dynamic viscosity of the mixed gas and metal phase, dynamic viscosity of the gas phase, and dynamic viscosity of the metal phase; P is the pressure; and $S_{mom}$ indicates any remaining momentum source terms, which include three components, as shown in Equation (9):(9)$$S_{mom}=S_{b}+S_{m}+\left(f_{sn}+f_{st}+P_{r}\right)\left|\nabla \alpha_{1}\right|$$
where $S_{b}$ is the buoyancy force, $S_{m}$ is the mushy zone drag force used to characterize the fluidity discrepancy induced by liquid–solid phase transition, $P_{r}$ is the recoil pressure due to metal evaporation, and $f_{sn}$ and $f_{st}$ are the surface tension and Marangoni effect at the interface between the liquid metal and gas, respectively. The term $\left|\nabla \alpha_{1}\right|$ is used to incorporate the surface forces into the volume forces.

The buoyancy force $S_{b}$ is considered using Boussinesq approximation [24], expressed as given in Equation (10):(10)$$S_{b}=\bar{\rho }g\beta \left(T-T_{ref}\right)$$
where g is the acceleration due to gravity, β refers to the thermal expansion related to the buoyancy force, T is the temperature, and T_ref is the reference temperature, typically set as the liquidus temperature. The drag force in the mushy zone $S_{m}\left(mushy\right)$ is calculated as follows [20]:(11)$$S_{m}\left(mushy\right)=-C\left[\frac{{\left(1-f_{l}\right)}^{2}}{{f_{l}}^{3}+C_{m}}\right]u$$
(12)$$f_{l}=\left\{\begin{array}{c} 0\;\;\;\;\;\;\;\;\;\;\;\;\;\;\;\;\;\;\;\;\;\;\;\;\;\;\;\;\;\;\;if\;T<T_{s}\\ \frac{T-T_{S}}{T_{L}-T_{S}}\;\;\;\;\;\;\;\;\;if\;T_{S}\leq T\leq T_{L}\\ 1\;\;\;\;\;\;\;\;\;\;\;\;\;\;\;\;\;\;\;\;\;\;\;\;\;\;\;\;\;\;\;if\;T>T_{L} \end{array}\right.$$
where *C* is a constant; its value is set to be large enough to ensure that the velocity decreases to zero when the local region fully solidifies. Typically, it is set to 105 or larger. $f_{l}$ is the liquid fraction of the metal phase, *T* is the temperature, and $T_{L}$ and $T_{S}$ represent the liquidus temperature and solidus temperature of the metal phase, respectively. $C_{m}$ is a custom small value used to avoid singularity in the mushy region during the calculation of the drag force. The recoil pressure $P_{r}$ acts normal to the local free surface, which is calculated as a function of the liquid surface temperature, defined as follows [21,22]:(13)$$S_{recoil}=0.54P_{a}exp\left(\frac{L_{v}M\left(T-T_{V}\right)}{RTT_{V}}\right)$$
where $P_{a}$ is the ambient pressure, $T_{V}$ is the vaporization temperature of the metal phase, $L_{v}$ is the latent heat of vaporization, M is the molar mass, and R is the universal gas constant. The surface tension $S_{m}\left(tension\right)$ can be obtained using a continuous surface force (CSF) model [23]:(14)$$f_{sn}=\sigma \kappa n$$
where σ is the surface tension coefficient. The Marangoni effect $f_{st}$ can be expressed as follows [23]:(15)$$S_{m}\left(Marangoni\right)=\frac{d\sigma }{dT}\left[\nabla T-\left(n\cdot \nabla T\right)n\right]$$
where the coefficient $\frac{d\sigma }{dT}$ represents the rate of variation in the surface tension with respect to temperature.

The energy conservation equation can be expressed as:(16)$$\frac{\partial \bar{\rho }\bar{c_{e}}T}{\partial t}+\nabla \cdot \left(\bar{\rho }u\bar{c_{e}}T\right)=\nabla \cdot \left(\bar{k}\nabla T\right)+S_{h}$$
where $\bar{c_{e}}$ is the equivalent specific heat capacity, and the calculation formula can be expressed as:(17)$$\bar{c_{e}}=\left\{\begin{array}{c} \alpha_{2}\left(c_{2}+\frac{L_{f}}{T_{L}-T_{S}}\right)+\alpha_{1}c_{1}\;\;\;\;\;\;\;\;\;\;\;\;\;\;T_{L}<T<T_{S}\\ \alpha_{1}c_{1}+\alpha_{2}c_{2}\;\;\;\;\;\;\;\;\;\;\;\;\;\;\;\;\;\;\;\;\;\;\;\;\;\;\;T\geq \;T_{L}\;or\;T\leq T_{S} \end{array}\right.$$
where $c_{1}$ and $c_{2}$ represent the specific heat of the gas phase and the metal phase, respectively; $L_{f}$ is the latent heat of fusion; and $\bar{k}$ is the thermal conductivity of the mixed gas and metal phase. The expression for $\bar{k}$ is:(18)$$\bar{k}=\alpha_{1}k_{1}+\alpha_{2}k_{2}$$
where $k_{1}$ and $k_{2}$ represent the thermal conductivities of the air phase and metal phase, respectively. The last term, $S_{h}$, represents the additional heat source terms applied to the surface of the molten film. It can be expressed as a combination of convective heat dissipation $S_{c}$, radiative heat dissipation $S_{r}$, vaporization heat dissipation $S_{v}$, and Gaussian beam heating $S_{l}$ [23]:(19)$$S_{h}=\left(S_{c}+S_{r}+S_{v}+S_{l}\right)\left|\nabla \alpha_{1}\right|$$
where the term $\left|\nabla \alpha_{1}\right|$ is used to incorporate the surface heat dissipation terms into the volume heat dissipation:(20)$$S_{c}=h_{c}(T-T_{a})$$
(21)$$S_{r}=k_{B}\varepsilon \left(T^{4}-T_{a}^{4}\right)$$
(22)$$S_{v}=-\varphi \frac{L_{v}M}{\sqrt{2\pi MRT}}P_{a}exp\left[\frac{L_{v}M\left(T-T_{v}\right)}{RTT_{v}}\right]$$
(23)$$S_{l}=\frac{2\eta P_{laser}}{\pi r^{2}}exp\left(-2\frac{{\left(z-z_{0}-vt\right)}^{2}+{\left(x-x_{0}\right)}^{2}}{R^{2}}\right)$$
Here, $h_{c}$ is the convective heat transfer coefficient; $T_{a}$ denotes the ambient temperature; $P_{a}$ is the atmospheric pressure; $k_{B}$ is the Stefan–Boltzmann constant; ε is the surface emissivity; φ is the evaporation coefficient, typically 0.82 [25]; $L_{v}$ is the latent heat of vaporization; $T_{v}$ is the vaporization temperature; η is the metal laser absorption coefficient; $P_{laser}$ is the laser power; r is the laser spot radius; x and z are the horizontal coordinates of the beam center during laser movement; $x_{0}$ and $z_{0}$ are the initial horizontal plane coordinates of the beam center; and v is the speed at which the laser moves along the *z*-axis.

### 3.2. Numerical Implementation

ANSYS 2022 R1 Fluent was used to simulate the molten pool during the powder-based DED-LB/M process in this work, which provides a suitable way to program corresponding user-defined functions for the temperature-dependent materials’ properties and source terms in the governing equations. The CT scan results, which indicated the flaw size distribution on an aluminum alloy component, are shown in Figure 2a. To facilitate the numerical computation, a specific location on the component was selected and simplified as a flat aluminum alloy plate with dimension of 14 × 7 × 2 mm, shown in Figure 2b. The air phase was also included in the numerical model and placed at the top of the metal plate to capture the interaction between the molten pool and the surrounding air. According to the CT-scanned flaw size, equivalent defects were created on the metal plate surface, with spherical diameters of 0.1, 0.5, and 0.9 mm and uniformly distributed along the deposit track. It is assumed that the required powder for deposition had already been delivered to the part surface before the high-energy laser beam was applied. The model considered a gravitational acceleration of 9.81 m/s^2^ (in the negative *y*-axis direction). The laser beam was assumed to irradiate the particle surface along the vertical direction (in the negative *y*-axis direction) and scan along the positive *z*-axis from the origin. The direct coupled DEM/FVM was employed to simulate the interaction between laser beam and powder particles. The DEM was used to calculate the initial positions and particle size information after particle delivery, shown in Figure 2c. This information was then transferred to the fluid computational domain as initialization data for the particle material, depicted in Figure 2d. The FVM was employed in the subsequent fluid dynamics calculations to capture the metal–air interface position, shown in Figure 2e.

The simulation of the powder delivery process was performed using the EDEM 2020 software. The physical shape of the delivered metal powder is defined as spherical, and their size distribution follows a normal distribution. The average particle size is 80 µm. The minimum particle size is 50 µm, and the maximum particle size is 110 µm. The material properties of both the metal particles and the metal substrate in the model include the Poisson ratio set to 0.334, the density set to 2700 kg/m^3^, and Young’s modulus set to 6.67 × 10^10^ Pa. The interaction coefficients between particle–particle and particle–substrate were defined as well. The restitution coefficient was set to 0.75, the static friction coefficient to 0.3, and the dynamic friction coefficient to 0.01. To simulate the powder delivery process, 6000 particles were first generated above the metal plate and then fell freely under gravity. The calculations revealed that the powder particles uniformly fell onto the metal plate surface, resulting in a powder thickness of 0.2 mm after 0.06 s, shown in Figure 3a. All spherical flaws on the surface were fully filled with the delivered powder particles. The majority of particles had velocities less than 6.53 × 10^−3^ m/s, indicating that the powder transport had been completed, depicted in Figure 3b.

The FVM simulation of the molten pool involves various aspects such as meshing of the computational domain, material selection, boundary condition and processing parameter settings, control of the numerical solver, etc. Regarding meshing, a fine mesh was adopted at the metal–air interface, where the molten pool is generated, shown in Figure 4. To ensure the feasibility of capturing essential physics in the laminar flow of the molten pool, the smallest mesh size was set to 0.02 mm while the largest mesh size was set to 0.8 mm. This took the minimal flaw size of 0.1 mm into consideration, as well as the largest geometry size of 14 mm. The tetrahedral element was adopted for the meshing, resulting in a total 14,549,170 elements in the model. The material parameters of the air and adopted aluminum alloy (AlSi10Mg) in the simulation are listed in Table 1 and Table 2. The processing parameters for the FVM model included a laser spot diameter of 3.5 mm, laser power $P_{laser}$ of 1600 W, scanning speed of 2160 mm/min, and scanning spacing of 1.2 mm. For the boundary conditions, the bottom and side surfaces of the metal plate were defined as adiabatic and non-slip walls. The top and side surfaces of the air were set as pressure outlets with a static pressure of 0. The loading of the heat and momentum source terms was implemented using Fluent’s User-Defined Function (UDF). The calculations employ a dynamic time step with an initial time step of 1 × 10^−8^ s, and the total physical time considered in the simulation is 0.24 s.

### 3.3. Result Analysis

The morphology of the molten pool and the evolution of the temperature field at different time points are illustrated in Figure 5. The computation reveals that surface flaws of 100 µm and 500 µm are successfully repaired. However, the surface flaw of 900 µm is not successfully repaired, resulting in a depression. The failure at the larger flaw size position is mainly due to an insufficient powder feed rate. A significant amount of unmelted particles and a rough surface are observed at the starting position of the laser scan. This can be attributed to inadequate energy absorption at the starting position, making it difficult to form a fully developed molten pool. The formation of the molten pool occurs around 0.01 s and stabilizes at around 0.2 s, forming a semi-ellipsoidal shape. Certain localized regions within the molten pool experience an excessive temperature gradient and fluid flow, which affects the deposition stability and leads to residual porosity within the deposit layer.

Figure 6 demonstrates the effects of increased powder layer thickness (or powder feed rate) to 0.4 mm on the deposition and repair outcomes. The results indicate that with the increased powder layer thickness, the surface flaws of various sizes disappear. However, this improvement comes at the cost of material loss on the substrate surface, leading to irreversible damage to the part dimensions. The thicker powder layer carries a larger amount of molten material along the scanning direction and eventually forms a hump, making it challenging to successfully repair surface flaws on the plate and even causing damage to the part. Therefore, it is crucial to use an appropriate powder feed rate and laser processing parameters to achieve a stable molten pool, smooth deposited layer, and effective repair of surface flaws.

## 4. Coupled Thermostructural Modeling of the Deposit Layers

### 4.1. Mathematical Modeling

The surface repair process using powder-based DED-LB/M refers to the first or second deposit layers. To compute thermal stress and deformation of the deposit track, a macroscale coupled thermostructural simulation is employed. The general form of the governing equation for the thermal conduction during the powder-based DED-LB/M process is presented as follows:(24)$$K\left(T\right)\cdot \left(\frac{\partial^{2}T}{\partial x^{2}}+\frac{\partial^{2}T}{\partial y^{2}}+\frac{\partial^{2}T}{\partial z^{2}}\right)+F=\frac{\partial H\left(T\right)}{\partial t}$$
where $K\left(T\right)$ is the temperature-dependent material’s thermal conductivity, T is the current temperature, F is the heat flux, t is time, and x, y, and z are the spatial directions. $H\left(T\right)$ is the temperature-dependent enthalpy, representing the latent heat evolution by phase transformation effect.

When $T\leq T_{s}$,
(25)$$H\left(T\right)=\rho \int_{T_{0}}^{T}C_{p}dT$$

When $T_{s}\leq T\leq T_{l}$,
(26)$$H\left(T\right)=\rho \int_{T_{0}}^{T_{s}}C_{p}dT+\rho L_{f}\left(\frac{T-T_{s}}{T_{l}-T_{s}}\right)$$

When $T>T_{l},$
(27)$$H\left(T\right)=\rho \int_{T_{0}}^{T_{s}}C_{p}dT+\rho L_{f}+\rho \int_{T_{l}}^{T}C_{p}dT$$
where $T_{l}$ is the liquidus temperature, $T_{s}$ is the solidus temperature, ρ is the density, $C_{p}$ is the specific heat, $L_{f}$ is the latent heat of melting, and $T_{0}$ is the room temperature, assumed to be 22 °C.

Structural analysis was used to compute residual stresses, which is generated by the strains corresponding to thermal expansion, contraction caused by temperature variation, and the nonelastic strains resulting from plastic deformation. The total strain increment vector can be expressed via the superposition of elastic, plastic, and thermal components, in the following form:(28)$$\Delta \varepsilon_{ij}^{tol}=\Delta \varepsilon_{ij}^{e}+\Delta \varepsilon_{ij}^{th}+\Delta \varepsilon_{ij}^{p}$$
where $\Delta \varepsilon_{ij}^{e}$ is the elastic strain increment, $\Delta \varepsilon_{ij}^{th}$ is the thermal strain increment, and $\Delta \varepsilon_{ij}^{p}$ is the plastic strain increment. The resulting stress increments are calculated via increments in elastic strain as follows:(29)$$\Delta \sigma_{ij}=D_{ijlm}\Delta \varepsilon_{ij}^{e}$$
where *E* is Young’s modulus, *v* is Poisson’s ratio, and $D_{ijlm}$ is the elastic stiffness tensor provided by Hook’s law:(30)$$D_{ijlm}=\frac{E}{1+v}\left[\frac{1}{2}\left(\delta_{il}\delta_{jm}+\delta_{lm}\delta_{ij}\right)+\frac{v}{1-2v}\delta_{ij}\delta_{lm}\right]$$
where δ is the Dirac function.

Combining Equation (27) with Equation (28) yields:(31)$$\Delta \sigma_{ij}=D_{ijlm}\left(\Delta \varepsilon_{ij}^{tol}-\Delta \varepsilon_{ij}^{th}-\Delta \varepsilon_{ij}^{p}\right)$$
where $\Delta \varepsilon_{ij}^{th}=\alpha \delta_{lm}\Delta T$, and α and ΔT are thermal expansion coefficient and temperature increment, respectively.

For the purpose of data mapping, the structural analysis utilizes the same mesh as the thermal analysis but with modified element and material properties. Material properties that vary with temperature are employed.

### 4.2. Numerical Implementation

ANSYS 2022 R1 Workbench was used to simulate the deposit layer during the powder-based DED-LB/M process in this work, which offers a suitable platform to perform both thermal and mechanical analysis and the data transfer between them. As depicted in Figure 7, the macroscale coupled thermostructural simulation employed the same metal plate, with a dimension of 14 × 7 × 2 mm, for consideration. The flaws with sizes of 0.1, 0.5, and 0.9 mm were incorporated on the plate’s surface as well. The actual metal deposit layer is assumed to be a finite element layer to ensure computational efficiency. The deposition process involved two layers, where deposit layers 1 and 2 were represented as finite element layers. The layers were incrementally added using the element birth and death method. A moving Gaussian heat source was applied for heat conduction in each finite element layer. The simulation incorporated direct coupling between thermal and structural mechanics, with the temperature field acting as a thermal load applied to subsequent structural analysis.

The establishment of this numerical model involves meshing, material property settings, boundary conditions, process parameters, and control of the computation process. The model employed mesh refinement at the locations of the surface defects on the metal plate to capture significant temperature and stress gradients. The minimum mesh size was set to 0.01 mm, while the maximum grid size was set to 0.4 mm. The computational domain was meshed using the tetrahedral elements, resulting in a total of 281,861 grid cells. Both the deposit material and the metal plate material are AlSi10Mg, and their thermodynamic material properties, varying with temperature, were obtained from the material library of ANSYS, shown in Figure 8.

The boundary conditions of the coupled thermostructural model were set so that the thermal convection with air was applied on the surface of the metal plate, and a fixed support was imposed on the bottom surface of the component. On the surfaces of each deposit layer, a Gaussian heat source moving in the positive *Z*-axis direction was defined using the Ansys Parametric Design Language (APDL) language. The processing parameters adopted were exactly the same as those of the coupled DEM-FVM model, where the diameter of the laser spot is 3.5 mm, the laser power is 1600 W, the scanning speed is 2160 mm/min, and the absorption coefficient of the metal material to laser energy is 0.35. For the thermal calculation, the initial temperature is set to 22 °C. The calculations are performed using a dynamic time step approach, with an initial time step of 3 × 10^−3^ s, a minimum time step of 3 × 10^−4^ s, and a maximum time step of 3 × 10^−2^ s.

### 4.3. Result Analysis

Figure 9 illustrates the temperature field variation during the powder-based DED-LB/M process. The analysis reveals that the maximum temperature gradually accumulated as each layer is deposited. The maximum temperature was concentrated at the scanning position of the laser beam, conforming to the Gaussian distribution characteristic of the energy. During the first deposit layer, the entire surface and interior of the deposit layer reached a temperature above the melting point, allowing for the formation of a sufficiently molten pool for repairing the component. The depth of the molten pool was sufficient to cover surface flaws of 0.1 mm and 0.5 mm, but it was insufficient to cover surface flaws with a depth of 0.9 mm. During the second deposit layer, the entire surface and interior of the deposit layer reached a temperature above the melting point, ensuring interlayer bonding.

Figure 10 illustrates the variation in the von Mises stress field during the powder-based DED-LB/M process. The analysis reveals that the maximum von Mises stress occurred at the bottom surface of the metal plate in the first deposit layer, measuring 236.86 MPa. As the second layer was deposited, the maximum von Mises stress decreased to 184.76 MPa, below the yield stress of the material, indicating that issues such as cracking are unlikely to occur during the deposition process. During the second deposit layer, the maximum von Mises stress appeared at the interfacial region between layers. No significant stress concentration was observed at the locations of the component’s flaws, demonstrating the feasibility of utilizing the powder-based DED-LB/M for surface repair of an aluminum alloy component.

The deformation field evolution during the powder-based DED-LB/M process is shown in Figure 11. The simulation results show that the maximum deformation gradually accumulated throughout the deposition process, and it reached its maximum value at the end of the second deposit layer (0.179 mm). The maximum deformation always occurred at the surface of the deposit layer and within the spot radius of the laser beam. This was attributed to the rapid temperature increase and thermal expansion of the material. No relative sliding resulting from deformation was observed at the surface flaw locations of the metal plate, indicating the reliability of the powder-based DED-LB/M process.

## 5. Conclusions

Both a mesoscale coupled DEM-FVM model of the molten pool and a macroscale coupled thermostructural model of the deposit layer for the powder-based DED-LB/M process were developed. The software ANSYS 2022 R1 Fluent and Workbench were employed to perform the mesoscale and macroscale simulations, respectively. The gradient grid size was applied to resolve tiny features for both the mesoscale (0.02 to 0.8 mm) and the macroscale models (0.01 to 0.4 mm). The computational time for the mesoscale coupled DEM-FVM model was 15 h, while that for the macroscale coupled thermostructural model was 12 h. The feasibility of repairing surface flaws using powder-based DED-LB/M technology was demonstrated, and the detailed conclusions are listed below:

(1)Micropores and bumping accompany the powder-based DED-LB/M process due to the extensive flow of the molten pool, and larger surface flaw sizes tend to result in an uneven deposit layer due to insufficient material supply. However, too much powder feed on the surface will lead to agglomeration of the molten materials along the scanning direction and severe damage of the metal base plate.(2)The maximum von Mises stress is far less than the yield stress of the adopted material, and no stress concentration exists during the powder-based DED-LB/M process. The total deformation will accumulate during the powder-based DED-LB/M process, and maximum deformation always occurs within the laser beam spot. No relative sliding phenomenon is observed between deposit layers.

The proposed multiscale model on the powder-based DED-LB/M for the surface repair of aluminum alloys would be necessary for a CAM engineer and machine operator to virtually validate and optimize the processing parameters before the actual manufacturing. Future work is expected to consider the mass loss caused by metal vaporization and the volume changes due to metal density variations to further increase the validity of the proposed mesoscale coupled DEM-FVM model of a molten pool. The thermal stresses during the surface repair of typical curved surfaces should be investigated using the established macroscale coupled thermostructural model. The effects of processing parameters such as laser absorptivity, laser scanning speed, and spot radius on the surface repair quality of the aluminum alloy component should also be studied. Simultaneously, a systematic experiment to verify the numerical model and reveal some new phenomena during the powder-based DED-LB/M process is expected. This experiment will likely use an in situ high-speed camera or X-ray imaging.

## Figures and Tables

**Figure 1 materials-17-03559-f001:**
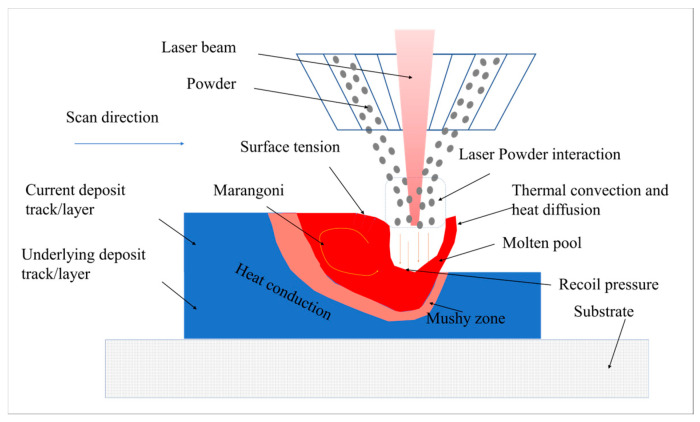
Physical processes of the laser-based direct energy deposition (DED-LB/M) process.

**Figure 2 materials-17-03559-f002:**
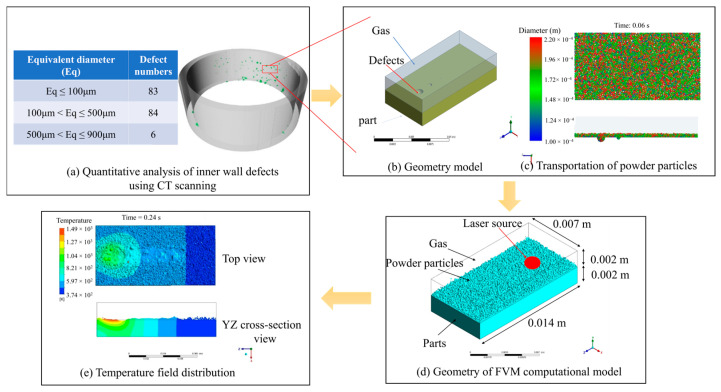
Roadmap for the coupled DEM-FVM modeling of the molten pool.

**Figure 3 materials-17-03559-f003:**
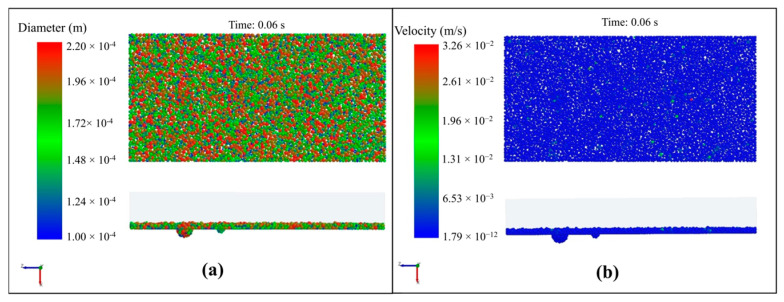
Simulation results of particle dynamics. (**a**) Particle positions and size distribution; (**b**) particle velocity distribution.

**Figure 4 materials-17-03559-f004:**
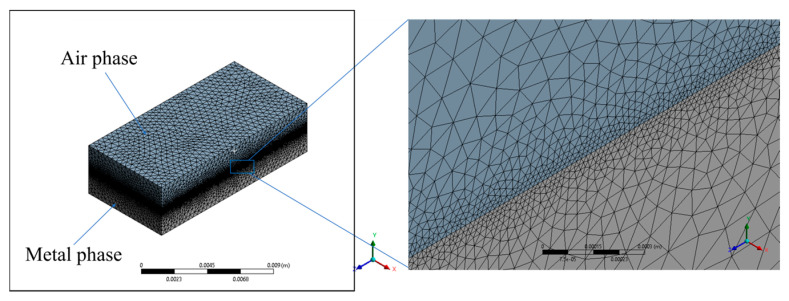
Meshing condition in the computational model.

**Figure 5 materials-17-03559-f005:**
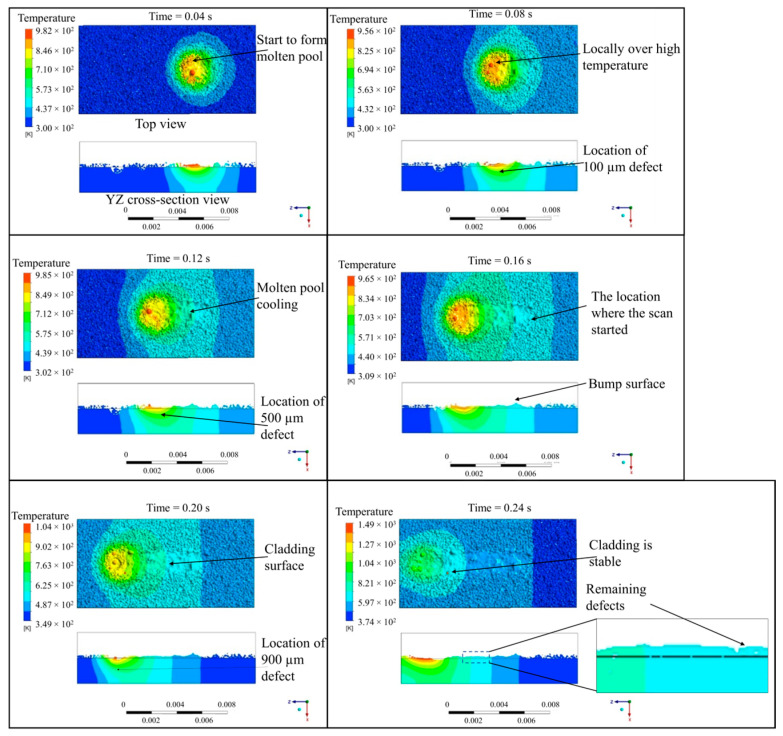
Morphology and temperature field evolution of the molten pool during the powder-based DED-LB/M process.

**Figure 6 materials-17-03559-f006:**
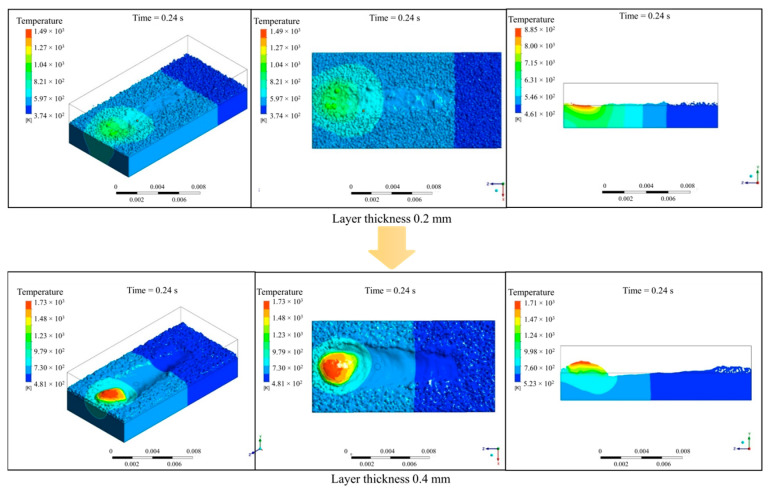
Comparison of repair qualities with different layer thicknesses or powder feed rates.

**Figure 7 materials-17-03559-f007:**
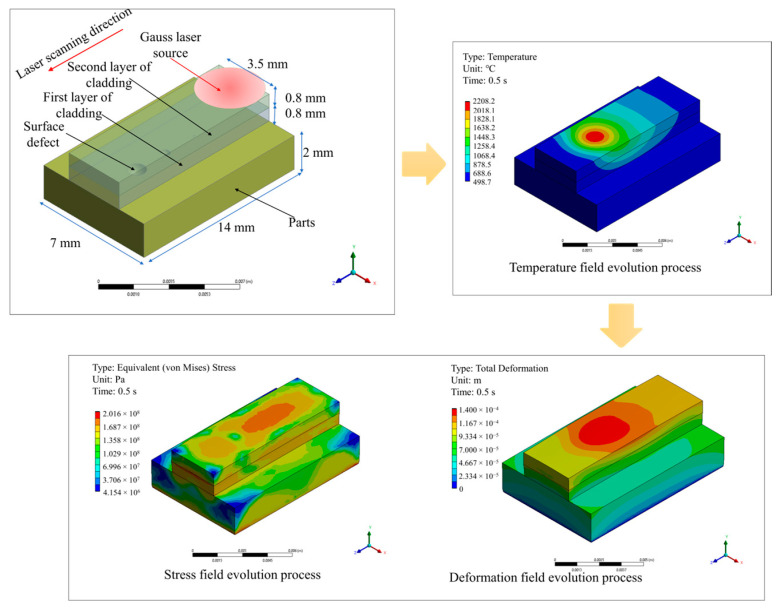
Roadmap of coupled thermostructural simulation of the powder-based DED-LB/M process.

**Figure 8 materials-17-03559-f008:**
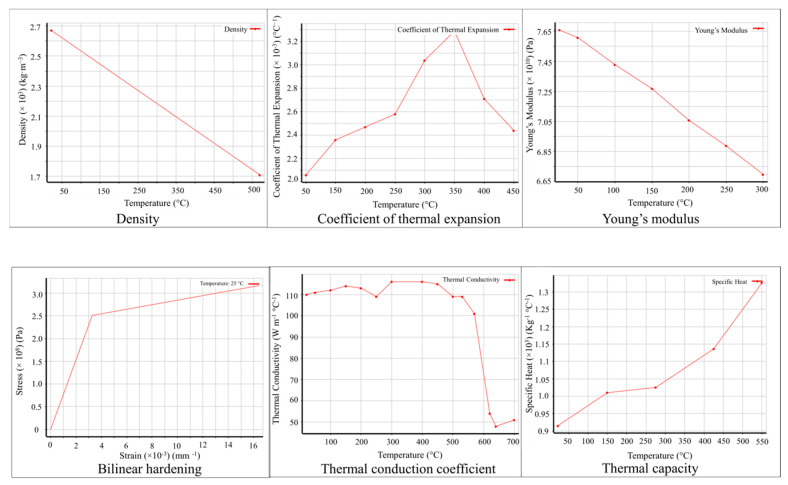
Temperature-dependent parameters of the aluminum alloy.

**Figure 9 materials-17-03559-f009:**
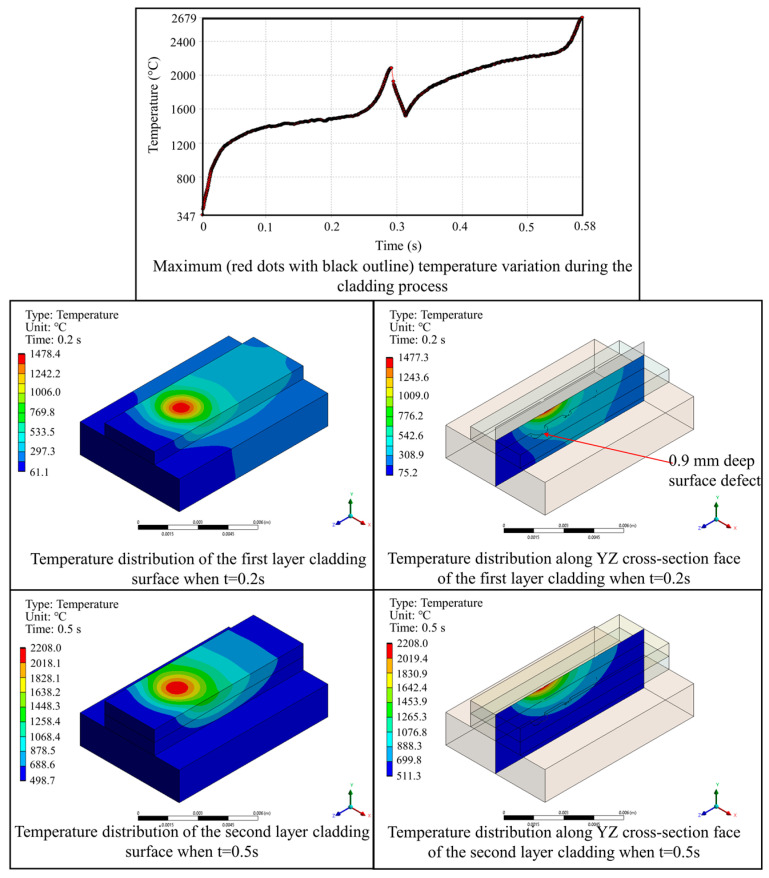
Evolution of temperature distribution during the powder-based DED-LB/M process.

**Figure 10 materials-17-03559-f010:**
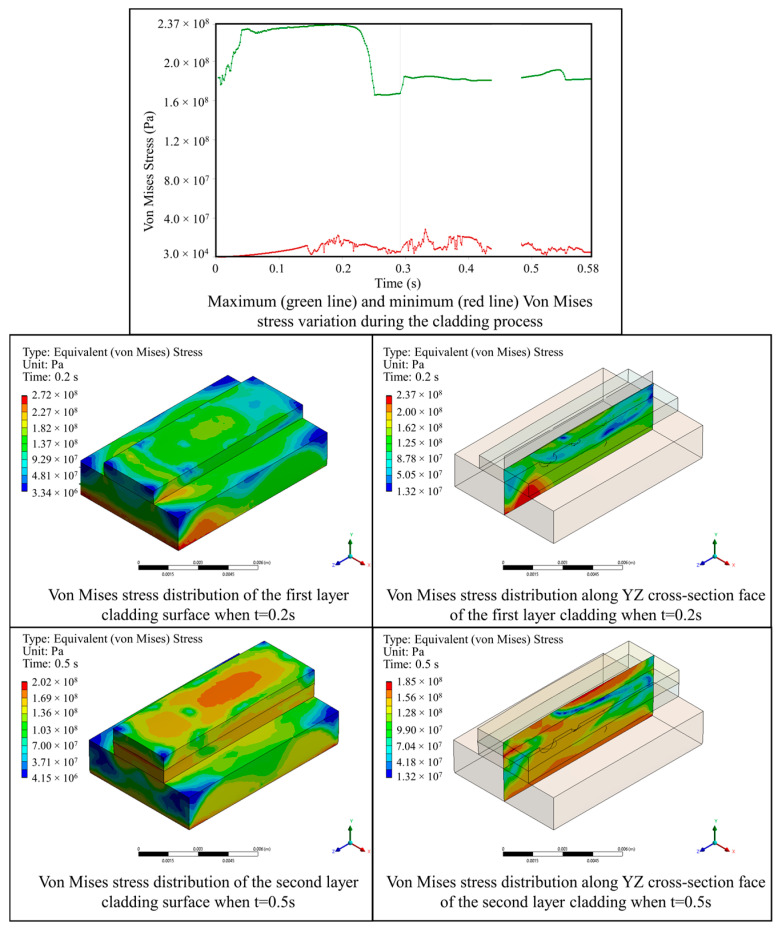
Evolution of von Mises stress distribution during the powder-based DED-LB/M process.

**Figure 11 materials-17-03559-f011:**
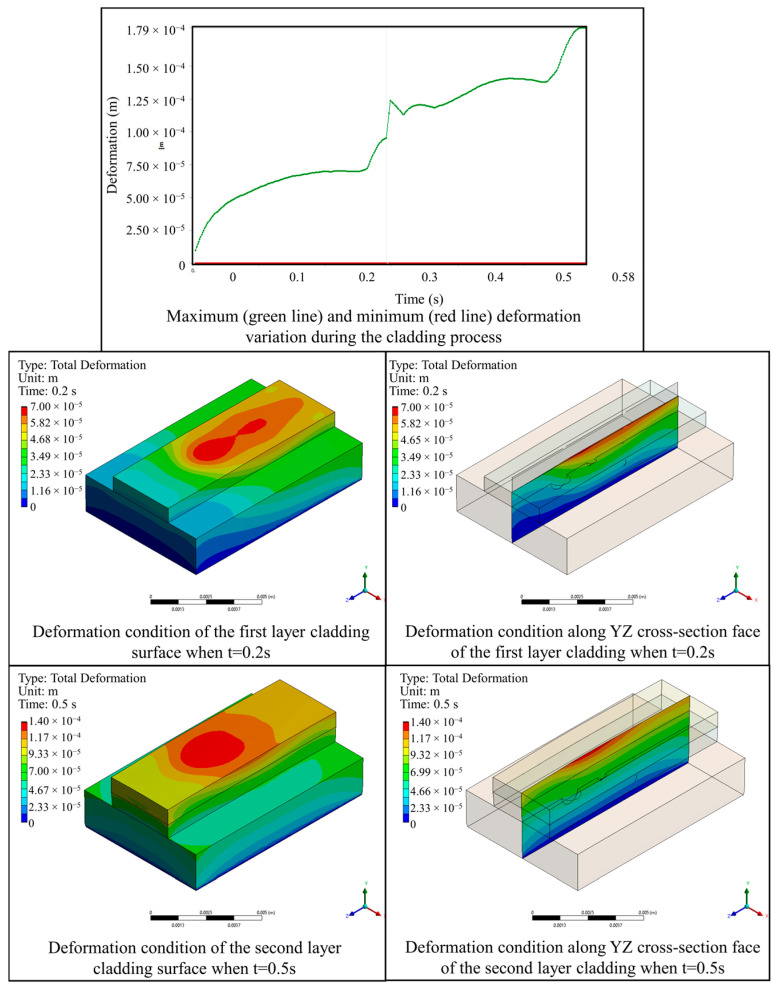
Evolution of deformation condition during the deposition process.

**Table 1 materials-17-03559-t001:** Material properties of air.

Symbol	Definition	Value
$\rho_{1}$	Density of gas phase (kg·m^−3^)	1.225
$c_{1}$	Specific heat of gas phase (J·kg^−1^·K^−1^)	1006.43
$k_{1}$	Thermal conductivity of gas phase (W·m^−1^·K^−1^)	0.0242
$\mu_{1}$	Dynamic viscosity of gas phase (kg·m^−1^·s^−1^)	1.7894 × 10^−5^

**Table 2 materials-17-03559-t002:** Material properties of aluminum alloy.

Symbol	Definition	Value
$T_{s}$	Solidus temperature (K)	890 [26]
$T_{L}$	Liquidus temperature (K)	929 [26]
$T_{v}$	Vaporization temperature (K)	2743 [26]
$\mu_{2}$	Liquidus dynamic viscosity (kg·m^−1^·s^−1^)	$0.00223exp\left(12200/8.3144T\right)$ [27]
$\rho_{2}$	Solidus density (kg·m^−3^)	2719
	Liquidus density (kg·m^−3^)	2828−0.3636T
$c_{2}$	Solidus specific heat (J·kg^−1^·K^−1^)	$798.85+0.3324T+9\times 10^{-5}T^{2}$ [26]
	Liquidus specific heat (J·kg^−1^·K^−1^)	1220 [26]
$k_{2}$	Solidus thermal conductivity (W·m^−1^·K^−1^)	$124.66+0.0561T+1\times 10^{-5}T^{2}$ [26]
	Liquidus thermal conductivity (W·m^−1^·K^−1^)	61 [26]
$L_{f}$	Latent heat of fusion (J·kg^−1^)	3.83 × 10^5^ [26]
$L_{v}$	Latent heat of vaporization (J·kg^−1^)	1.087 × 10^7^ [26]
$h_{c}$	Convective heat transfer coefficient (W·m^2^·K)	10
σ	Surface tension coefficient (kg·s ^−2^)	$0.914-0.00035\left(T-890\right)$ [28]
R	Universal gas constant (J·mol^−1^·K^−1^)	8.314
$k_{B}$	Stefan–Boltzmann constant (W·m^−2^·K^−4^)	5.67 × 10^−8^
η	Laser beam absorptivity	0.35 [29]
M	Molar mass (kg·mol^−1^)	0.026982
*ε*	Surface emissivity	0.3

## Data Availability

Due to privacy concerns, the ANSYS 2022 R1 Fluent UDF code presented in this study is only available on request from the corresponding author.

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
