# Peer review of "Multiscale Simulation of Laser-Based Direct Energy Deposition (DED-LB/M) Using Powder Feedstock for Surface Repair of Aluminum Alloy"

_materials, 2024, doi:10.3390/ma17143559_

Round 1
Reviewer 1 Report
Comments and Suggestions for Authors
Dear Authors,
Thank you for this very interesting work related to the topic of developing simulation tools for laser based Directed Energy Deposition and aluminum as working material. Please find hereafter some comments that must be addressed during the revision phase.
Abstract & Title
· You should follow the standards as it regards the name of the process. In this case, the name should be Laser based Directed Energy Deposition (DED- LB/M) and be used throughout the manuscript.
· You have to clarify the type of feedstock in both title and abstract since the defects that you are mentioning (pores, uneven deposition layer) are caused by different mechanisms each time. In the powder based DED-LB/M, the unequal powder particle size may cause the unstable flow and uneven material flow rate per unit time, while this is not the case for wire DED-LB/M which has consistent material flow etc.
· In this section, please avoid including introductory material. In page 2 (lines 47-48) the text looks very similar to the abstract (lines 16-17)
Introduction
Line 50-53: The authors’ statement it is not completely true. The authors could consult the following study from literature, where a monitoring set up has captured the heat field either directly or indirectly via tracking the evolution of melt pool dimensions.
· Panagiotis Stavropoulos, Georgios Pastras, Konstantinos Tzimanis, Nikolas Bourlesas, Addressing the challenge of process stability control in wire DED-LB/M process, CIRP Annals, 2024,ISSN 0007-8506, https://doi.org/10.1016/j.cirp.2024.04.021.
Section 3- Coupled DEM-FVM Modeling of the Molten Pool
At the beginning of this section, the authors introduce the simplification of their modelling approach. They should quantify the effect of each assumption. For this purpose, can use sources from literature or by conducting such simulation.
What is the purpose of Fig 3-6? Should we identify similarities of the process output with the outputs of the simulation tools.
Section 4: Coupled Thermo-Structural Modelling of the Deposit Layers
In DED modelling, many of the difficulties of simulation comes from the design characteristics of the part. Do the authors consider different design shapes to investigate? If no, how these outputs can be reflected in real industrial scenarios?
The output of the simulation tool can be used from the machine operator, the CAM engineer or the CAD engineer? How the outputs can be used in every day bases?
Conclusions
What is the computational time and the computational resources, grid size etc.
References
Please add the following reference for reasons of completeness:
Stavropoulos, P., & Foteinopoulos, P. (2018). Modelling of additive manufacturing processes: a review and classification. Manufacturing Review, 5, 2.
Comments on the Quality of English LanguageMinor editing of English language required
Reviewer 2 Report
Comments and Suggestions for Authors
1. Why is ANSYS chosen for simulation? Other software such as COMSOL is also good at multiphysics fluid flow simulation.
2. Line 273 to 275, is there any mesh sensitivity analysis conducted to show that results are convergent? 3. Table 3-2, could you be more specific which property is referred from which reference? 4. Resolution is low for figures such as 4-1 and 4-4, and it is difficult to see the details. 5. Is Fig 4-2 from any literature or material library of ANSYS? 6. Line 284, the physical time simulated is only .24 s. Are you able to simulate the whole process within this short period of time? This is also making experiment validation almost impossible. 7. Since there is no experimental validation, could you add a session of future work and discuss what simplifications and assumptions are made in the simulation that should be addressed in the future?
Round 2
Reviewer 1 Report
Comments and Suggestions for Authors
The authors have addressed the comments and the paper can be considered for publication
Reviewer 2 Report
Comments and Suggestions for Authors
Comments are addressed properly.